# Prototypical Variational Autoencoder for Few-shot 3D Point Cloud Object Detection

**Weiliang Tang**[*]
The Chinese University of Hong Kong
wltang21@cse.cuhk.edu.hk

**Biqi Yang**[*]
The Chinese University of Hong Kong
bqyang@cse.cuhk.edu.hk

**Xianzhi Li**
Huazhong University of Science and Technology
xzli@hust.edu.cn

**Pheng-Ann Heng**
The Chinese University of Hong Kong
pheng@cse.cuhk.edu.hk

**Yunhui Liu**
The Chinese University of Hong Kong
yhliu@mae.cuhk.edu.hk

**Chi-Wing Fu**
Department of CSE and SHIAE
The Chinese University of Hong Kong
cwfu@cse.cuhk.edu.hk

## Abstract

Few-Shot 3D Point Cloud Object Detection (FS3D) is a challenging task, aiming to detect 3D objects of novel classes using only limited annotated samples for training. Considering that the detection performance highly relies on the quality of the latent features, we design a VAE-based prototype learning scheme, named prototypical VAE (P-VAE), to learn a probabilistic latent space for enhancing the diversity and distinctiveness of the sampled features. The network encodes a multi-center GMM-like posterior, in which each distribution centers at a prototype. For regularization, P-VAE incorporates a reconstruction task to preserve geometric information. To adopt P-VAE for the 3D object detection framework, we formulate Geometric-informative Prototypical VAE (GP-VAE) to handle varying geometric components and Class-specific Prototypical VAE (CP-VAE) to handle varying object categories. In the first stage, we harness GP-VAE to aid feature extraction from the input scene. In the second stage, we cluster the geometric-informative features into per-instance features and use CP-VAE to refine each instance feature with category-level guidance. Experimental results show the top performance of our approach over the state of the arts on two FS3D benchmarks. Quantitative ablations and qualitative prototype analysis further demonstrate that our probabilistic modeling can significantly boost prototype learning for FS3D.

## 1  Introduction

3D point cloud object detection is a fundamental scene understanding task, aiming at recognizing and localizing objects in point-based scenes. Recent deep-learning methods are mostly fully supervised, requiring extensive labeled data to support the model training. Such a data-hungry setting is, however,

[*] Equal contributions to the works. This work was supported by Shenzhen Portion of Shenzhen-Hong Kong Science and Technology Innovation Cooperation Zone under HZQB-KCZYB-20200089. This work was supported by the Hong Kong Centre for Logistics Robotics, the Research Grants Council-General Research Fund (No. 14201620). This work was supported by InnoHK of the Government of Hong Kong via the Hong Kong Centre for Logistics Robotics. This work was supported by project MMT-p2-21 of the Shun Hing Institute of Advanced Engineering (SHIAE).

37th Conference on Neural Information Processing Systems (NeurIPS 2023).

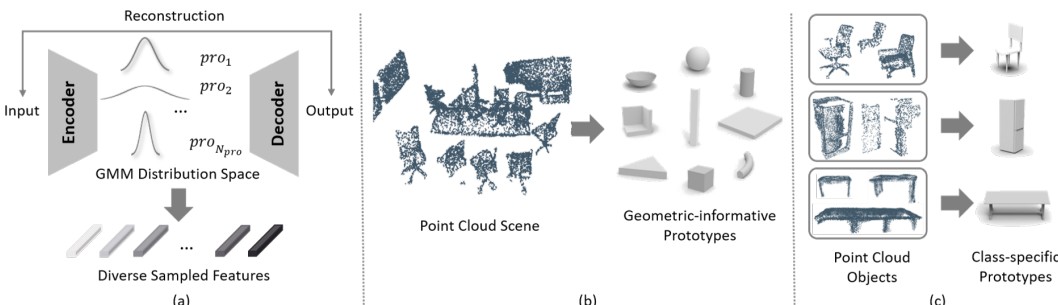

Figure 1: (a) The basic structure of our proposed Prototypical VAE, where we learn a GMM-like distribution for feature sampling that can be regularized by 3D reconstruction tasks. (b) At the scene level, each point cloud scene can be reconstructed by composing local geometric components, we call them geometric-informative prototypes. (c) At the object level, objects of the same category can be represented in a unified fashion by a class prototype, named the class-specific prototype.

impractical in real applications, considering the heavy and costly loads to densely annotate the data. Inspired by humans' capability of recognizing novel objects by learning from limited data, Few-shot Learning (FSL) [1, 2, 3, 4, 5] has been proposed to enable neural networks to handle novel data by training on large labeled data of base classes and small labeled data of novel classes.

Prototype learning is an effective approach to achieve FSL [2, 6, 7, 8, 9, 10, 11]. It has been applied successfully to various 2D vision tasks, e.g., object detection [12, 13, 14], semantic segmentation [15, 16, 17, 18, 19, 20], and instance segmentation [21, 22]. In prototype learning, some principles are first defined to separate the prototypes, e.g., by category [23, 17, 13, 14, 16, 21], by partial similarity [19, 11], or by intrinsic characteristics [12, 15, 17, 18, 20]. These prototypes can be viewed as clustering centers, such that we can group the latent features based on our pre-defined principles. Importantly, the prototypes provide transferable knowledge that allows us to generalize a model for handling novel classes, thereby facilitating representation learning for downstream tasks.

In this work, we exploit prototype learning for Few-Shot 3D Point Cloud Object Detection, named FS3D. We aim to predict accurate 3D bounding boxes on objects of novel classes, given only a small amount of labeled novel-class exemplars. To address FS3D with prototype learning, Zhao et al. [10] propose a pioneering work. Yet, their learning scheme lacks fine-level supervision, as the intermediate features are simply averaged to update the prototypes, which are then used to augment features for sequential detection. There are two fundamental issues. First, without strong regulation on prototype learning, the prototypes can lose substantial 3D information and become less geometric-informative. Second, due to the data imbalance, prototypes of novel classes are clearly underrepresented than those of the base classes, so the latent space can easily overfit the base ones.

To address the above issues, we propose a variational autoencoder approach particularly designed for prototype learning, named Prototypical VAE (abbr. P-VAE). As Fig. 1(a) illustrates, instead of directly learning the features, our P-VAE learns the distribution parameters, by which we can construct a Gaussian Mixture Model(GMM)-based posterior for sampling features from the probabilistic latent space. For issue 1, we propose to perform reconstruction tasks with an encoder-decoder architecture in our P-VAE framework to better preserve the geometric information in the intermediate features. For issue 2, our P-VAE allows us to sample diverse and varied features controlled by a multi-center distribution. Hence, even with limited novel samples, our approach can still gather richer information to generate good-quality representative prototypes for the novel classes.

To enhance the object detection task with the proposed P-VAE, we formulate two crucial extensions: Geometric-informative Prototypical VAE (GP-VAE) and Class-specific Prototypical VAE (CP-VAE). As Fig. 1 shows, we can view real-world 3D objects from two aspects. In a broader sense, all the objects, regardless of their class labels, can be reasoned by a set of local geometric components; see Fig. 1(b). For GP-VAE, each component is a geometric-informative prototype, such that a given point cloud scene can be constructed by combining these prototypes. Narrowing the scope to each object, objects of the same category have similar shapes, while objects of different categories look more different; see Fig. 1(c). Hence, we propose CP-VAE to obtain class-specific prototypes, such that each prototype can serve as a strong shape prior to guiding the object-level feature learning. We incorporate the two modules in our object detection pipeline, leveraging GP-VAE to account for scene-level feature extraction and CP-VAE for object-level feature refinement; see Fig. 2.

We conduct extensive experiments on two FS3D benchmarks, FS-ScanNet and FS-SUNRGB [10]. The results in Sec. 4.2 show that our method significantly outperforms the SOTA approaches in various few-shot settings. More analysis on prototypes in Sec. 4.3 demonstrate the effectiveness of leveraging VAE for prototypical learning. To summarize, our contributions are listed as follows:

- We propose the Prototypical VAE (P-VAE) to learn multiple distributions centering at prototypes, such that it can construct a probabilistic latent space that allows us to effectively sample diverse features with higher controllability and flexibility.
- To harness P-VAE for object detection, we extend it for geometric-informative and class-specific prototypical VAEs (i.e., GP-VAE & CP-VAE) based on our observations.
- Results demonstrate the superiority of our new approach for FS3D and further studies analyze the prototypes and explain the effectiveness in detail; see Sec. 4 and Appendix.

## 2 Related Works

**Variational Autoencoder (VAE)** is first introduced by Kingma et al. [24] to learn a generative model for constructing a distribution-based latent space. It provides a probabilistic representation learning scheme with theoretical support. Bornschein et al. [25] assume that features follow a multi-center prior distribution, which encourages unsupervised deep clustering in the embedding space. Norouzi et al. [26] select exemplars from the training data and use them to model the embedding space with GMM. To strengthen the robustness, [27, 28, 29] balance between reconstruction authenticity and generation diversity; [30, 31, 32] aims at preventing VAE models from collapse; [33, 34, 35] explore usages of different regularizations. VAE is widely adopted in 3D generation tasks, e.g., point cloud generation [36, 37, 38] and scene reconstruction [39, 40]. However, it has not been fully explored for 3D perception tasks, especially for the challenging FS3D task.

**Few-shot 2D Image Object Detection (FS2D)** methods can be classified into two categories. The first family aims to address the data imbalance problem, for example, by fine-tuning with balanced data [41], augmenting objects by different scales [42], augmenting novel samples with generative networks [43], learning reasonable and unbiased latent space [44], etc. The second family focuses on learning support (base)-novel (query) correlations. Kang et al. [45] reweight the query features given the support features; Han et al. [46] fuse the support and query features by spatial alignment; Fan et al. [47] and Yan et al. [48] exploit deep support information to guide the region proposal network and the detection heads; some other works incorporate support information learned from the backbone to guide the feature extraction of the query images, using transformers [49, 50, 51, 52, 53] or graph convolution networks [49]. An intuitive solution is to adopt these FS2D strategies to tackle FS3D, as some of our comparison methods in Sec. 4.2. However, their detection results (see Tab. 1 and 2) are far worse than our P-VAE. The reason is that 3D point clouds are intrinsically more disordered and complex than pixel-grid 2D images, thus naturally lacking abundant data with full annotations.

**FSL for 3D Point Cloud** is under-explored, as we discussed before. As a pioneer work, Fan et al. [54] tackle the one-shot point cloud classification, exploring base-novel sample relations with graph convolution. Cheraghian et al. [55] introduce a transductive zero-shot learning method with a triplet loss to obtain a better classifier. Zhao et al. [11] incorporate FSL for 3D semantic segmentation with a transformer for multi-prototype transductive inference. To our best knowledge, Prototypical VoteNet [10] is the first and only work on FS3D that harnesses prototype learning. However, their experimental results, refer to the indoor scenes in Fig. 3, are not satisfactory. First, it lacks a strong regulation for prototype learning, like our reconstruction task, to preserve latent geometric information. Second, the quality of their novel-class prototypes is impaired due to data imbalance, whereas we propose a new VAE-based approach to sample features with significant diversity.

## 3 Method

In this section, we first define the FS3D task and give a modular overview of our pipeline in Sec. 3.1. Then, we present the formulation of our novel Prototypical VAE (P-VAE) to learn a probabilistic-distribution-based feature space in Sec. 3.2. Building upon the P-VAE, we develop (i) Geometric-informative Prototypical VAE (GP-VAE) (Sec 3.3) and (ii) Class-specific Prototypical VAE (CP-VAE) (Sec 3.4), which are customized for learning prototypes of varying geometric components and varying object categories, respectively.

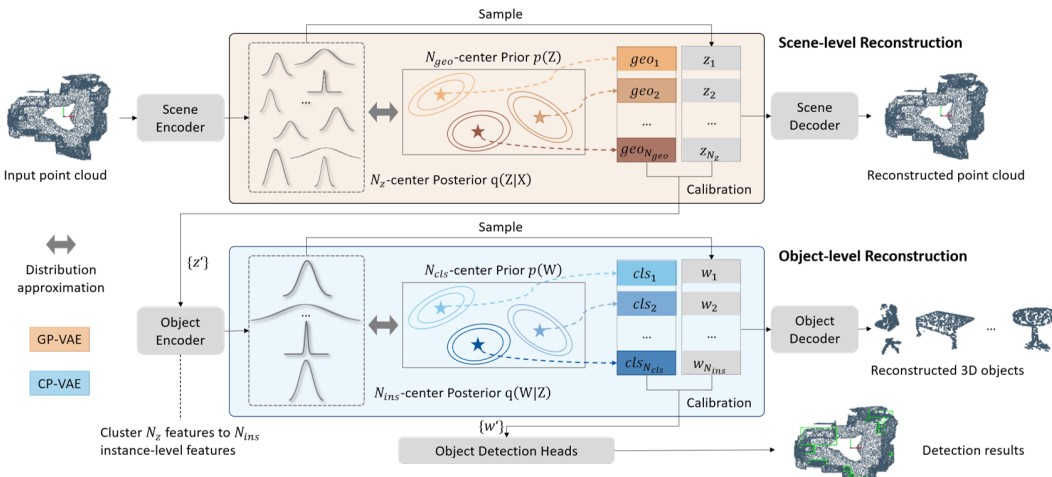

Figure 2: Illustration of our pipeline. We incorporate two crucial tasks to facilitate the framework to learn representative prototypes: a scene-level reconstruction task with GP-VAE and an object-level reconstruction task with CP-VAE.

## 3.1 Problem Definition and Framework Overview

In the FS3D task, the full class set $\mathbb{C}$ is split into a base set $\mathbb{C}_{base}$ and a novel set $\mathbb{C}_{novel}$, where $\mathbb{C}_{base} \cap \mathbb{C}_{novel} = \emptyset$. Each base class in $\mathbb{C}_{base}$ has plenty of annotated objects, while the amount of labeled samples is very limited for each novel class in $\mathbb{C}_{novel}$. The FS3D annotation for an instance of class-$i$ is $(pos, dim, c_i)$, denoting the box center point, 3D box dimensions, and the object class label. Following the FSL definition, the $K$-shot setting means only $K$ annotated objects can be used for training in each novel class. Our goal is to enable a model pre-trained on sufficiently-annotated $\mathbb{C}_{base}$ samples to generalize over new classes using only $K$ labeled objects per class in $\mathbb{C}_{novel}$.

Fig. 2 shows the overall pipeline. Built upon the detection framework, we perform two reconstruction tasks during the training to regulate the prototype learning and model the probabilistic latent space:

**(i) Scene-level reconstruction,** in which we incorporate our proposed GP-VAE to learn geometric prototypes $\{geo\}$ with a variational encoder-decoder architecture. From the learnt distribution, we sample features $\{z\}$ for decoding and further refine $\{z\}$ to $\{z'\}$ with the geometric prototypes for down-stream object detection (see Sec. 3.3); and

**(ii) Object-level reconstruction,** in which we adopt a voting-based object encoder to cluster the scene-level features $\{z'\}$ into object-level features and incorporate the proposed CP-VAE to learn class-specific prototypes $\{cls\}$ as distribution centers. Similar to (i), we sample features $\{w\}$ for decoding and output calibrated features $\{w'\}$ (see Sec. 3.4).

In the end, we use multiple feed-forward network heads, which are fed with $\{w'\}$, to generate instance-level 3D box positions and dimensions as the detection results. During the inference, we can skip the reconstruction decoders and adopt the pre-trained prototypes for forward data flow.

## 3.2 Prototypical Variational Autoencoder

Our proposed Prototypical VAE (P-VAE) is built upon a standard VAE as its backbone to learn high-level prototypes. Referring to Fig. 1(a), the overall architecture consists of an encoder and a decoder. $X$ is the variable that represents our data and the encoded latent space follows a certain prior distribution $p(Z)$. The encoder $f$ parameterized by $\theta_f$ takes the point cloud $x \in \mathbb{R}^{N_x \times 3}$ as the input and generates parameters $\mu \in \mathbb{R}^{N_z \times d}$ and $\sigma \in \mathbb{R}^{N_z \times d}$, where $N_z$ is the number of parameters $\{\mu_i, \sigma_i\}$ and $d$ is the latent space dimension. We adopt them to obtain a set of predicted posteriors $q_{\theta_f}(Z_i|X) \sim \mathcal{N}(\mu_i, \sigma_i)$, where $i = 1...N_z$.

Based on these $N_z$ normal distributions, we can sample $N_z$ new features $\{z_i | z_i \in \mathbb{R}^d\}_{i=1}^{N_z}$. Then, we adopt a decoder $g$, which is parameterized as $\theta_g$, taking each $z_i$ to reconstruct a point cloud with $N_p$ points $p_i \in \mathbb{R}^{N_p \times 3}$. Finally, we combine these reconstruction results to form our point cloud prediction $p$. Given the ground-truth point cloud $\hat{p}$, the optimization goal for P-VAE is

$$\underset{\theta_f, \theta_g}{\arg\min}[\text{CD}(p, \hat{p}) + \frac{1}{N_z} \sum_{i=1}^{N_z} \text{KL}(q_{\theta_f}(Z_i|X) \| p(Z))], \tag{1}$$

where CD denotes the Chamfer Distance that supervises the reconstruction task and KL denotes the Kullback-Leibler Divergence to regularize the posteriors $q_{\theta_f}(Z_i|X), i = 1...N_z$ to stay close to the latent space prior $p(Z)$. By far, we can learn this distribution space in a standard VAE term.

Importantly, we focus on obtaining prototypes based on latent distributions. Denoting $\{pro_i|pro_i \in \mathbb{R}^d\}_{i=1}^{N_{pro}}$ as $N_{pro}$ prototypes, our goal is to use them as clustering centers to group the latent embeddings. Hence, we define $p(Z)$ following an approximate GMM, which is composed of $N_{pro}$ independent normal distributions that each centers at $pro_i$ and has variance $I$:

$$p(Z = z) = \sum_{i=1}^{N_{pro}} \frac{exp(-\frac{1}{2}\|z - pro_i\|_2^2)}{A(2\pi)^{d/2}} \mathbb{1}\{i = \arg\min_i \|z - pro_i\|_2\}. \tag{2}$$

Compared with the standard GMM, our approximate GMM is designed with two strategies. First, we adopt a fixed weight $\frac{1}{A}$ ($\int_{\mathbb{R}^d} p(Z = z)dz = 1$) instead of distinctive learned weights for these normal distributions, such that each prototype can be equally accessed. This design is for imitating a real-world situation, where all the geometric components share the same probability to be observed among common geometric structures. Second, we adopt a hard regularization $\mathbb{1}$ in Eq. (2), such that each sampled feature $z_i$ can be directly assigned to its nearest prototype. In this way, our intermediate features are more organized, since they are discriminatively grouped by the prototypes. By introducing $pro$, we modify the posterior distribution $q_{\theta_f}(Z_i = z\|X) = A\exp(-\frac{1}{2}\|z - \mu_i\|_2^2)\mathbb{1}\{\|z - pro_j\|_2^2 \leq \|z - pro_k\|_2^2\}$, where $A$ is the normalization term and $pro_j$ is the closest prototype to $\mu_i$.

With Eq. (2), we can optimize the KL term of Eq. (1) and rewrite Eq. (1) as

$$\arg\min_{\theta_f, \theta_g}[\text{CD}(p, \hat{p}) + \frac{1}{2}(\sum_{i=1}^{N_{pro}}[\|\sigma_i\|_2^2 - \|\log\sigma_i^2\|_1] + \underbrace{\arg\min_{pro}[\sum_{i=1}^{N_{pro}} \sum_{\mu_j \in B_i} \|pro_i - \mu_j\|_2^2])}_{\text{Term Pro}}], \tag{3}$$

where $B_i = \{\mu_k|\|\mu_k - pro_i\|_2 \leq \|\mu_k - pro_j\|_2, j = 1...N_{pro}\}$, indicating the set of features that are assigned to prototype $pro_i$ by least Euclidean distance. Please see Appendix for detailed derivations.

To find the optimal prototypes $\{pro_i\}_{i=1}^{N_{pro}}$ is to solve the 'Term Pro' in Eq. (3), which can be viewed as a clustering task to seek $N_{pro}$ clustering centers, such that the mean distance between all pairs of the sampled features to their closest prototypes is minimized. We adopt the K-means algorithm updated with Exponential Moving Average (EMA) to enhance the training efficiency. Overall, at the $t^{th}$ training iteration, $pro_i$ is estimated as

$$pro_i^{t+1} \leftarrow \alpha pro_i^t + (1 - \alpha)\frac{\sum_{\mu_k \in B_i} \mu_k}{|B_i|}, B_i = \{\mu_k|\|\mu_k - pro_i\|_2 \leq \|\mu_k - pro_j\|_2, \forall j = 1, \ldots, N_{pro}\}, \tag{4}$$

where $\alpha$ is the smoothness hyperparameter and all the prototypes are initialized with the Gaussian random initialization before the training. After we obtain the new prototypes in an iteration, we take them back to Eq. (3) to get our loss for training the network:

$$\mathcal{L} = \text{CD}(p, \hat{p}) + \frac{1}{2}(\sum_{i=1}^{N_{pro}}[\|\sigma_i\|_2^2 - \|\log\sigma_i^2\|_1] + \sum_{i=1}^{N_{pro}} \sum_{\mu_j \in B_i} \|pro_i - \mu_j\|_2^2), \tag{5}$$

where the first term is the reconstruction loss and the last term is the clustering loss for encouraging each sampled feature to stay close to its associated nearest prototype.

### 3.3 Scene Reconstruction with Geometric Prototypical VAE (GP-VAE)

The detection framework starts by extracting features from the input scene. Here, we incorporate GP-VAE to enhance the feature quality. We reconstruct the whole point cloud scene with GP-VAE, and obtain geometric-informative prototypes denoted as $\{geo_i\}_{i=1}^{N_{geo}}$. These prototypes represent basic geometric components of common real-world 3D objects.

Please refer to Fig. 2 for GP-VAE. We use the PointNet++ [56] as our encoder $f_{scene}$. It takes a raw point cloud $x \in \mathbb{R}^{N_x \times 3}$ as input and outputs parameters $\{\mu_i, \sigma_i\}_{i=1}^{N_z}$ to construct the posterior normal distributions $\mathcal{N}(\mu_i, \sigma_i), i = 1...N_z$. We sample $\{z_i\}_{i=1}^{N_z}$ from the learned distributions and build a decoder $g_{scene}$ to reconstruct the original point cloud scene. For $g_{scene}$, we adopt the architecture

of FoldingNet [57], which transforms a plane into a 3D object surface. Specifically, we use furthest point sampling (FPS) to obtain $N_p$ points on a $0.4m \times 0.4m$ plane $B \in \mathbb{R}^{N_p \times 2}$. For each $z_i$, we repeat the feature $N_p$ times then concatenate them with plane $B$, followed by two 2-layer MLPs to generate $p_i \in \mathbb{R}^{N_p \times 3}$. We use Eq. (4) to obtain the geometric prototypes $\{geo_i\}_{i=1}^{N_{geo}}$. We combine the $N_z$ predicted point clouds to construct the scene-level result $p \in \mathbb{R}^{N_z N_p \times 3}$, then use Eq. (5) to supervise the scene-level reconstruction, where the ground truth is the input point cloud $x$.

Instead of directly using the sampled features $\{z_i\}_{i=1}^{N_z}$ to cluster the per-object features, we calibrate $\{z_i\}_{i=1}^{N_z}$ with the geometric prototypes via a cross attention layer. This calibration enriches the information diversity of each feature by not only its nearest prototype but also other geometric prototypes. For brevity, we use $geo \in \mathbb{R}^{N_{geo} \times d}$ as the united representation of the prototypes:

$$z_i' = \text{Softmax}(\frac{\mathbf{Q}(z_i)\mathbf{K}^\top(geo)}{\sqrt{d}})\mathbf{V}(geo), \tag{6}$$

where $\mathbf{Q}$, $\mathbf{K}$, and $\mathbf{V}$ are three MLPs to generate the query, key, and value in the attention framework. In this way, we can obtain robust embedding features $\{z_i'\}_{i=1}^{N_z}$.

### 3.4 Object Reconstruction with Class-specific Prototypical VAE

Given the features $\{z_i'\}_{i=1}^{N_z}$ generated in Sec. 3.3, we use CP-VAE to reconstruct each object and obtain class-specific prototypes as shown in Fig. 1(c). We build a voting-based [58] encoder $f_{object}$. For detection purpose, we need per-object features $\{\hat{z}_i\}_{i=1}^{N_{ins}}$, where $N_{ins}$ is the number of the predicted instances. Therefore, we use a 1-layer MLP to predict the feature offsets and point offsets from $\{z_i'\}_{i=1}^{N_z}$, then adopt FPS on the shifted points to obtain $N_{ins}$ votes. For each vote, we collect a maximum number of 16 shifted features within 0.3m from the vote, then use max pooling on these features to generate the associated object-level feature $\hat{z}_i$. Another built MLP is then fed with $\{\hat{z}_i\}_{i=1}^{N_{ins}}$ to output parameters $\{\mu_i, \sigma_i\}_{i=1}^{N_{ins}}$, which can be used to construct the posterior truncated normal distributions $\mathcal{N}(\mu_i, \sigma_i)$ where $i = 1...N_{ins}$. We sample $\{w_i\}_{i=1}^{N_{ins}}$ and build a FoldingNet decoder $g_{object}$ (Sec. 3.3) for object-level reconstruction. For each $w_i$, $g_{object}$ outputs an unique 3D object $p_i$.

According to the sampling and grouping mechanisms in feature extraction, we reversely map each $w_i$ to a set of corresponding points in $x$ (we will detail this operation in the Appendix), which constructs our reconstruction target $\hat{p}_i$. Therefore, we modify the reconstruction term of Eq. (5) to

$$\mathcal{L} = \frac{1}{N_{ins}} \sum_{i=1}^{N_{ins}} \text{CD}(p_i, \hat{p}_i) + \frac{1}{2N_{cls}}(\sum_{i=1}^{N_{cls}}[\|\sigma_i\|_2^2 - \|\log \sigma_i^2\|_1] + \sum_{i=1}^{N_{cls}} \sum_{\mu_j \in B_i} \|cls_i - \mu_j\|_2^2), \tag{7}$$

where $B_i = \{\mu_k | \|\mu_k - cls_i\|_2 \leq \|\mu_k - cls_j\|_2, j = 1...N_{cls}\}$. Importantly, we further modify the prototype update formula in Eq. (4) to customize for class-specific prototype learning. As discussed before, we can reversely map each $\mu_i$ to a set of input points. During the training, we use this point set to find the nearest ground-truth instance by the least center Euclidean distance, then assign the corresponding category label to $\mu_i$. We collect features of the same category to update the class-specific prototypes accordingly. Denoting $l(\cdot)$ as the operation to get the category index for each feature, to obtain precise and convincing class-specific prototypes, Eq. (4) is rewritten as

$$cls_i^{t+1} \leftarrow \alpha cls_i^t + (1 - \alpha)\frac{\sum_{\mu_j \in B_i} \mu_j}{|B_i|}, \quad B_i = \{\mu_j | l(\mu_j) = i\}. \tag{8}$$

Similar to Sec. 3.3, we refine the sample features $\{w_i\}_{i=1}^{N_{ins}}$ with the prototypes by using a cross-attention augmentation, where we adopt $cls \in \mathbb{R}^{N_{cls} \times d}$ as the united representation of the prototypes:

$$w_i' = \text{Softmax}(\frac{\mathbf{Q}(w_i)\mathbf{K}^\top(cls)}{\sqrt{d}})\mathbf{V}(cls). \tag{9}$$

## 4 Experiment

### 4.1 Experiment Setup

We conduct experiments on two benchmarks **FS-ScanNet** and **FS-SUNRGBD** [10]. The **FS-ScanNet** dataset consists of 18 object categories and 1,513 point cloud scenes in total. This benchmark has two

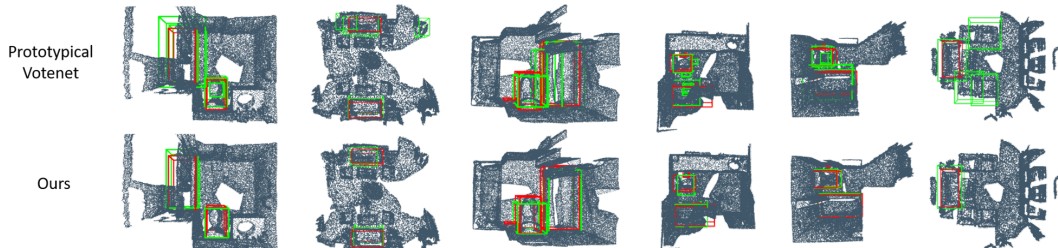

Figure 3: Example detection results by our method and Prototypical VoteNet [10] (SOTA) on **FS-ScanNet split-1** with $K = 5$. The red bounding boxes are the ground truths, while the green bounding boxes are the predictions.

different base/novel splits, each including 6 novel classes and 12 base classes. The **FS-SUNRGBD** dataset contains 5,000 point cloud scenes, covering 10 object categories. It has only one base/novel split, including 4 novel classes and 6 base classes. Following the standard 3D object detection protocol, comparison results on both datasets are quantitatively evaluated on mean Average Precision (mAP) under IoU thresholds 0.25 and 0.5, denoted as $AP_{25}$ and $AP_{50}$. More details on the splits of both datasets, the network architecture, and the training scheme can be found in the Appendix.

| Method | split-1 | | | | | | split-2 | | | | | |
|---|---|---|---|---|---|---|---|---|---|---|---|---|
| | 1-shot | | 3-shot | | 5-shot | | 1-shot | | 3-shot | | 5-shot | |
| | $AP_{25}$ | $AP_{50}$ | $AP_{25}$ | $AP_{50}$ | $AP_{25}$ | $AP_{50}$ | $AP_{25}$ | $AP_{50}$ | $AP_{25}$ | $AP_{50}$ | $AP_{25}$ | $AP_{50}$ |
| Beseline [58] | 11.72 | 8.02 | 21.13 | 9.57 | 28.63 | 15.69 | 8.79 | 1.71 | 18.19 | 5.52 | 22.68 | 11.64 |
| Generalized FS3D [59] | 12.03 | 8.19 | 24.90 | 10.26 | 29.29 | 16.67 | 9.19 | 1.87 | 19.41 | 6.80 | 25.18 | 12.74 |
| PointContrast-VoteNet [60] | 12.59 | 8.52 | 20.12 | 11.16 | 25.83 | 15.49 | 9.55 | 1.97 | 18.44 | 5.23 | 20.06 | 10.19 |
| Fractal-VoteNet [61] | 11.81 | 7.57 | 21.38 | 10.11 | 24.66 | 14.73 | 9.16 | 1.68 | 15.65 | 4.88 | 20.35 | 10.26 |
| Meta-Det3D [62] | 10.28 | 4.03 | 23.42 | 10.64 | 25.65 | 13.88 | 5.21 | 1.32 | 15.44 | 4.37 | 22.13 | 7.09 |
| Prototypical VoteNet [10] | 15.34 | 8.25 | 31.25 | 16.01 | 32.25 | 19.52 | 11.01 | 2.21 | 21.14 | 8.39 | 28.52 | 12.35 |
| **Ours** | **16.00** | **10.22** | **31.60** | **19.37** | **32.84** | **22.39** | **12.66** | **4.15** | **21.27** | **10.09** | **31.70** | **14.43** |

Table 1: Comparing the FS3D performance of our method against existing works on **FS-ScanNet**.

## 4.2 Comparison Results

**Comparison Approaches.** We quantitatively compare our method with VoteNet [58] baseline and five recent methods that utilize different FSL techniques to tackle FS3D. For the baseline, we train the VoteNet detector and then directly infer its performance on the novel test data without FSL. The other five recent methods are (i) Generalized FS3D [59], an incremental fine-tuning method with a sample adaptive balance loss built upon TFA [41]; (ii) PointContrast-VoteNet [60], which obtains generic and transferable features by self-supervised pre-training; (iii) Fractal-VoteNet [61], which adopts a pre-training similar to (ii), whereas it learns features from fractal geometry 3D patterns (this concept can be viewed as a simplified geometric prototype without any regulation principle); (iv) Meta-Det3D [62], which uses a meta detector [63] to output class-specific re-weighted vectors to refine RoI features (this is similar to our CP-VAE but their class vectors are negatively affected by data imbalance); (v) Prototypical VoteNet [10], the most recent SOTA approach that groups the latent features to obtain prototypes.

| Method | 1-shot | | 2-shot | | 3-shot | | 4-shot | | 5-shot | |
|---|---|---|---|---|---|---|---|---|---|---|
| | $AP_{25}$ | $AP_{50}$ | $AP_{25}$ | $AP_{50}$ | $AP_{25}$ | $AP_{50}$ | $AP_{25}$ | $AP_{50}$ | $AP_{25}$ | $AP_{50}$ |
| Baseline [58] | 5.46 | 0.22 | 6.52 | 0.77 | 13.73 | 2.20 | 20.47 | 4.50 | 22.99 | 5.90 |
| Generalized FS3D [59] | 6.81 | 1.58 | 12.21 | 2.02 | 17.52 | 4.69 | 22.12 | 5.97 | 22.84 | 6.76 |
| PointContrast-VoteNet [60] | 7.03 | 1.17 | 8.16 | 2.33 | 20.32 | 4.19 | 21.13 | 4.49 | 21.03 | 6.71 |
| Fractal-VoteNet [61] | 7.54 | 1.39 | 9.16 | 3.01 | 21.08 | 4.25 | 21.23 | 5.68 | 22.01 | 6.77 |
| Meta-Det3D [62] | 6.77 | 0.73 | 8.29 | 1.21 | 15.37 | 2.99 | 19.60 | 4.67 | 24.22 | 5.68 |
| Prototypical VoteNet [10] | 12.39 | 1.52 | 14.54 | 3.05 | 21.51 | 6.13 | 24.78 | 7.17 | 29.95 | 8.16 |
| **Ours** | **14.36** | **2.42** | **22.28** | **4.30** | **27.70** | **8.73** | **31.55** | **13.84** | **33.21** | **13.98** |

Table 2: Comparing the FS3D performance of our method against existing works on **FS-SUNRGBD**.

**FS-ScanNet Benchmark.** Tab. 1 summarizes our quantitative comparison results on **FS-ScanNet**. Despite their achievements in FS2D, the improvements of fine-tuning and pre-training [59, 60, 61] over the baseline are marginal for FS3D. They rely on a robust feature space pre-trained with large-scale datasets (ImageNet1K [64] for 2D), which are hard to obtain and annotate for the 3D scenes.

As we discussed before, [61] and [62] use the concept of prototypes but their learning schemes are less in-depth. Our proposed P-VAE consistently outperforms all the above methods, thanks to the proposed distribution-based prototype learning scheme adopted in GP-VAE and CP-VAE. Fig. 3 shows convincing qualitative results compared with SOTA [10], our predictions (green) better overlap with the ground truths (red), while [10] may produce more false positives and box errors.

**FS-SUNRGBD Benchmark.** Tab. 2 summarizes our quantitative comparison results on **FS-SUNGRBD**. Compared with **FS-ScanNet**, **FS-SUNGRBD** is more challenging with higher clutter levels and more severe object diversity. Under such cases, our method still consistently surpasses SOTA [10] in all the few-shot settings, especially on the more strict $AP_{50}$ metric.

**Comparison on Base Classes.** We further test only on base classes to see if our prototype learning scheme for novel samples may impair the features learned on the base samples. Please see Tab. 3. VoteNet (Baseline) is specifically designed for base classes. Compared with it, Prototypical VoteNet (Proto) suffers from

| Method | FS-ScanNet-1 | | FS-ScanNet-2 | | FS-SUNRGBD | |
|---|---|---|---|---|---|---|
| | $AP_{25}$ | $AP_{50}$ | $AP_{25}$ | $AP_{50}$ | $AP_{25}$ | $AP_{50}$ |
| Baseline [58] | 57.96 | 32.60 | 54.63 | 35.76 | 47.77 | 26.78 |
| Proto [10] | 53.83 | 28.20 | 51.22 | 32.41 | 46.07 | 25.26 |
| Ours | **58.16** | **32.69** | **56.05** | **38.77** | **47.92** | **27.75** |

Table 3: Comparing base-class results on three datasets.

consistent precision drops, which means the extracted knowledge from the novel samples is less incompatible with one of the base classes, thus the base feature space is hurt by those chaotic signals. Our P-VAE is able to better preserve the base feature structures and improves the origin detection performance on all the benchmarks. This observation demonstrates that our novel and base features are better aligned and can be simultaneously enhanced by our distribution-based prototype learning.

## 4.3 Ablation Studies and Detailed Analysis

| Method | 3-shot | | 5-shot | |
|---|---|---|---|---|
| | $AP_{25}$ | $AP_{50}$ | $AP_{25}$ | $AP_{50}$ |
| Baseline | 21.13 | 9.57 | 28.63 | 15.69 |
| + GP-VAE | 29.69 | 15.18 | 33.48 | 22.18 |
| + CP-VAE | 27.11 | 18.88 | 31.92 | 21.91 |
| + GP-VAE & CP-VAE | **31.60** | **19.37** | **32.84** | **22.39** |

Table 4: Ablating our major components.

| Module | PL scheme | 3-shot | | 5-shot | |
|---|---|---|---|---|---|
| | | $AP_{25}$ | $AP_{50}$ | $AP_{25}$ | $AP_{50}$ |
| GP-VAE | proto | 25.71 | 13.57 | 29.04 | 17.17 |
| | AE | 29.57 | 14.48 | 33.14 | 19.16 |
| | VAE | **29.69** | **15.18** | **33.48** | **22.18** |
| CP-VAE | proto | 26.14 | 17.43 | 31.13 | 19.16 |
| | AE | 26.56 | 17.98 | 31.61 | 21.45 |
| | VAE | **27.11** | **18.88** | **31.92** | **21.91** |

Table 5: Analyzing different PL schemes.

**Module Ablation.** We ablate the major components of our framework on **FS-ScanNet split-1** with $K = 3, 5$. From Tab. 4, we can see that each component contributes to improving the performance and our full pipeline attains the highest ratings for all settings. We can observe that individual usage of CP-VAE works less effectively than individual usage of GP-VAE. The reason is that we incorporate GP-VAE in the scene-level reconstruction, which is prior to the object-level reconstruction by using CP-VAE. If we omit GP-VAE, the features from the scene encoder are not sufficiently representative to support the sequential clustering, thus it could hurt prototype learning in CP-VAE.

**Prototype Learning Schemes.** Based on the Prototypical VoteNet [10], denoted as Proto in Tab. 5, we optimize the prototype learning scheme to AE and VAE. AE introduces a reconstruction task with an encoder-decoder architecture, while the latent features to optimize prototypes are directly learned from the encoder instead of sampled from the distributions. AE can be viewed as non-probabilistic prototypical learning. VAE is built upon AE by further constructing a probabilistic latent space to enable us to sample diverse features. It can be viewed as probabilistic prototypical learning. As Tab. 5 reports, we conduct independent quantitative comparisons for GP-VAE and CP-VAE on **FS-ScanNet split-1** with $K = 3, 5$. The results of AE over Proto show that incorporating a reconstruction task helps preserve the geometric information within the latent embeddings. VAE further consistently outperforms AE on all the metrics for both GP-VAE and CP-VAE, demonstrating that the probabilistic modeling can significantly boost feature learning for FS3D.

**Prototype Quality.** Thanks to the reconstruction tasks, we can visualize our prototypes as point clouds. Fig. 4(left) shows some of the geometric-informative prototypes learned by the GP-VAE; more results can be found in the Appendix. From them we can see the diversity and distinctiveness of the prototypes, each exhibiting a certain geometric pattern. In Fig. 4(right), we illustrate our class-specific prototypes learned by the CP-VAE. These results show certain class-level abstractions of the general object appearance.

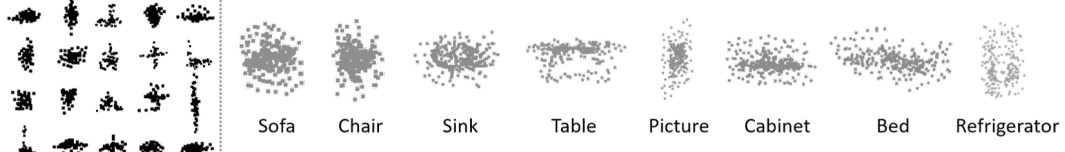

Sofa  Chair  Sink  Table  Picture  Cabinet  Bed  Refrigerator

Figure 4: Left: reconstructed examples of learned geometric-informative prototypes, 49 points for each. Right: reconstructed examples of learned class-specific prototypes, 256 points for each.

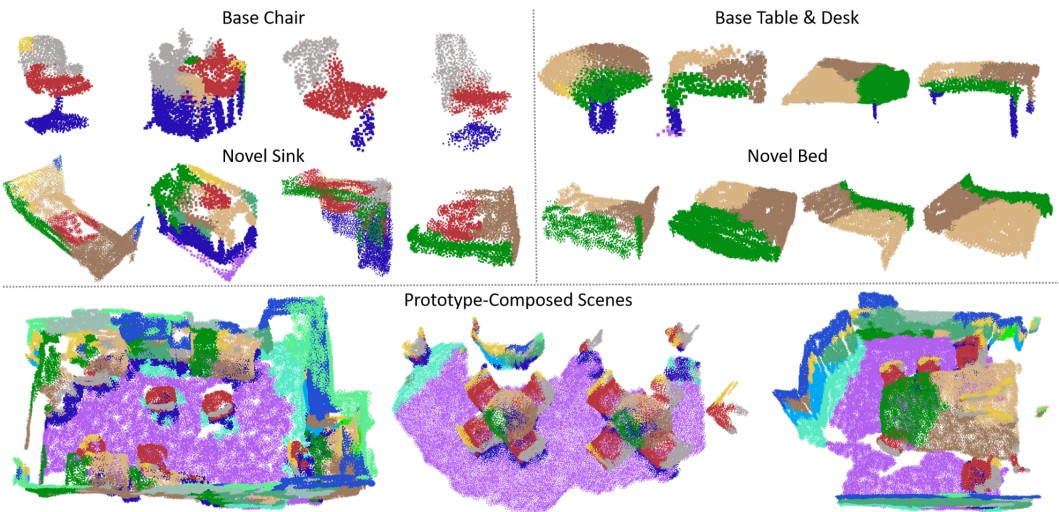

Base Chair  Base Table & Desk

Novel Sink  Novel Bed

Prototype-Composed Scenes

Figure 5: Illustration of geometric-informative prototypes compositing objects and scenes. Each color represents a unique prototype learned from GP-VAE. These prototypes show strong inter-class physics consistency and intra-class semantic correspondences.

**Prototype Effects.** After generating the prototypes with a well-designed prototype learning scheme, our final question is: do these prototypes help representation learning?

For geometric-informative prototypes, we illustrate how these prototypes help to construct an object or a whole scene in Fig. 5, where each color represents a geometric-informative prototype. We can observe that each object, regardless of its base/novel category label, is composed of multiple prototypes with semantic correspondence patterns. For example, the red prototype indicates an open semicircle shape, appearing not only in the base chair class (consistently at the seat region) but also in the novel sink class (consistently at

| Num. of Geo-Proto | $AP_{25}$ | $AP_{50}$ |
|---|---|---|
| $N_{geo} = 16$ | 22.19 | 6.48 |
| $N_{geo} = 32$ | 28.59 | 14.52 |
| $N_{geo} = 64$ | 31.78 | 18.51 |
| $N_{geo} = 128$ | 32.84 | 22.39 |
| $N_{geo} = 256$ | 32.06 | 20.30 |

Table 6: Ablation for $N_{geo}$.

the center region). Results demonstrate that these prototypes are robust geometric patterns that help enhance the generalization ability for both base and novel classes, so they can provide transferable and consistent 3D hints for scene-level feature encoding. Please see Tab. 6 for quantitative and Appendix for qualitative ablations on varying numbers of prototypes, denoting $N_{geo}$ in Sec. 3.3.

For class-specific prototypes, in Fig. 6, we use t-SNE visualization to show the high-level object embeddings $\{w_i\}_{i=1}^{N_{ins}}$ of five common categories. From left to right: without class-specific prototypes, CP-AE (defined in Tab. 5), and CP-VAE. Compared with the vanilla feature space, where the features are disordered and chaotic, incorporating class prototypes leads to a more compact latent space, such that each prototype can be more separated and the sampled features are better organized by clustering.

**Pluggable P-VAE.** Besides the VoteNet baseline, we extend multiple point cloud detection frameworks with our P-VAE. Please refer to Appendix for quantitative results that demonstrate the effectiveness of our P-VAE as a plug-in module for various networks to better adapt FS3D.

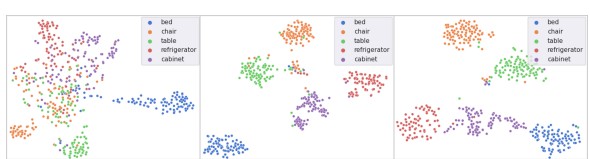

Figure 6: Visualize object-level embeddings with t-SNE.

## 5 Conclusions

To tackle few-shot 3D point cloud object detection (FS3D), we design a specific VAE for prototype learning, named Prototypical VAE (P-VAE). P-VAE constructs a probabilistic latent space that allows us to sample diverse and distinctive features, such that we can learn more representative prototypes even if the novel samples are limited. To adopt P-VAE for FS3D, we propose the Geometric-informative GP-VAE and the Class-specific CP-VAE based on our common observations in the real world. Experimental results demonstrate the effectiveness of P-VAE, and further prototype analysis shows its potential value in other prototype-based few-shot 3D perception tasks.

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
