# Appendix

## A  Theoretical Derivation of P-VAE

This section details the derivation from Eq. 1 to Eq. 3.

Let $P(Z) = \frac{1}{A} \sum_{i=1}^{N_{pro}} p_i(Z)$, where

$$p_i(Z = z) = \frac{\exp(-\frac{1}{2}\|z - pro_i\|_2^2)}{(2\pi)^{d/2}} \mathbb{1}\{i = \arg\min_i \|z - pro_i\|_2\}, \tag{10}$$

and $q_{\theta_f}(Z_i = z|X) = B\exp(-\frac{1}{2}\|z - \mu_i\|_2^2)\mathbb{1}\{\|z - pro_i\|_2^2 \leq \|z - pro_j\|_2^2\}$. Where $B$ is the normalization term, and $pro_i$ is the $pro$ with minimum distance to $\mu_i$. Let $q'_{\theta_f}(Z|X) = B\exp(-\frac{1}{2}\|z - \mu_i\|_2^2)$, and $p'_i(Z) = \frac{\exp(-\frac{1}{2}\|z - pro_i\|_2^2)}{(2\pi)^{d/2}}$ Then we have:

$$
\begin{aligned}
&\mathrm{CD}(p, \hat{p}) + \mathrm{KL}(q_{\theta_f}(Z|X)\|p(Z)) \\
&= \mathrm{CD}(p, \hat{p})) + \mathrm{KL}(q_{\theta_f i}(Z|X)\|p_i(Z)) \\
&\leq \mathrm{CD}(p, \hat{p})) + \mathrm{KL}(q'_{\theta_f}(Z|X)\|p'_i(Z)) \\
&= \mathrm{CD}(p, \hat{p}) + \frac{1}{2}(\|\sigma_i\|_2^2 - \|\log \sigma_i^2\|_1 - d) + \frac{1}{2}\|pro_i - \mu_i\|_2^2)
\end{aligned}
\tag{11}
$$

Let $B_i = \{\mu_k | \|\mu_k - pro_i\|_2 \leq \|\mu_k - pro_j\|_2, \forall j = 1, \ldots, N_{pro}\}$. Our optimization target for multiple $q_{\theta_f}$ becomes:

$$\arg\min_{\theta_f, \theta_g}[\mathrm{CD}(p, \hat{p}) + \frac{1}{2}(\sum_{i=1}^{N_{pro}}[\|\sigma_i\|_2^2 - \|\log \sigma_i^2\|_1] + \arg\min_{pro}[\sum_{i=1}^{N_{pro}} \sum_{\mu_j \in B_i} \|pro_i - \mu_j\|_2^2])]. \tag{12}$$

## B  Experimental Details

### B.1  Network Details and Training Schemes

Based on Sec. 3, our PointNet++ [3]-like scene encoder $f_{scene}$ has four set abstraction layers that downsample input points to 2048, 1024, 512, and 256 with an increasing radius of 0.2, 0.4, 0.8, and 1.2, and three feature propagation layers that upsample points to 512, 1024, and 2048.

For both GP-VAE and CP-VAE, the number of attention heads is empirically set to 4. We customize a fixed weight 0.2 to the KL divergence such that we can bias more towards the reconstruction loss in Eq. (5) and Eq. (7). We set the smoothness coefficient, denoted as $\alpha$ in Eq. (4) and Eq. (8) for updating $cls$ and $geo$, to 0.999. Particularly for GP-VAE, the number of geometric-informative prototypes $N_{geo}$ is set to 128, following the statistics in [1]; the number of parameter sets $N_z$ is set to 2048; the number of each reconstructed point cloud $N_p$ is 49 (i.e., a $7 \times 7$ grid in FoldNet). For CP-VAE, the number of class-specific prototypes $N_{cls}$ depends on the dataset class statistics; the number of parameter sets $N_{ins}$ is set to 256; the point number of each reconstructed object is 128. For VAE evidence lower bound, we multiply 0.1 to KL divergences to bias more towards reconstruction loss.

The network is fed with 40K points for **FS-ScanNet** and 20K points for **FS-SUNRGBD**, followed by a set of data augmentation techniques including random flipping, random rotation, and random scaling as proposed in [4]. For network training, we adopt the AdamW [5] optimizer to run 36 epochs with a weight decay of 0.01. The initial learning rate is set as 0.008 and 0.001 for **FS-ScaneNet** and **FS-SUNRGBD** respectively. For fast convergence, we do not predict $\sigma$ and set $\sigma$ to 0 for the posterior distributions of both GP-VAE and CP-VAE in the first 20 epochs (in Eq. (5) and Eq. (7)).

## B.2 Point-Feature Mapping

For CP-VAE (please refer to Sec. 3.4), we mention that we will reversely map each object-level feature $w_i$ to a set of original points. This part may bring doubts for readers so next we will introduce it in detail.

Gathering and grouping are widely-used operations for feature extraction in PointNet++-based networks. This is the basis for our point-feature mapping. Illustrated in Fig. 7, in GP-VAE, we downsample the input points and group them to extract $\{z\}$. Then in CP-VAE, we cluster $\{z'\}$ to obtain $\{w\}$ by voting. Reversely, we can map $\{w\}$ to $\{z\}$, and then map $\{z\}$ to the corresponding points in $x$. By the above procedures, for each object-level feature $w_i$ we obtain a point set, in which each point has a category label. To assign one label for $w_i$ among a bunch of different labels, there can be various principles to find the closest object instance, e.g., by 3D box overlap ratio, by CD distance, and by center Euclidean distance. In CP-VAE we choose the center Euclidean distance, and we also conduct an ablation experiment for principle choosing. Please refer to Tab. 7, the experiment is conducted on **FS-ScanNet Split-1** 5-shot. Although the result shows only marginal improvements, we practically find that using center Euclidean distance can help the network converge much faster.

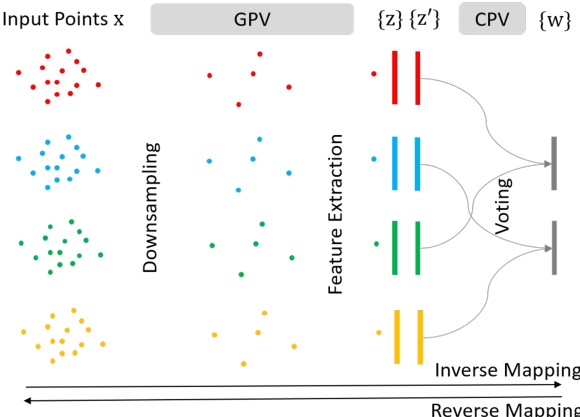

Figure 7: Forward data flow of extracting features from the input scene.

| **Principles** | $AP_{25}$ | $AP_{50}$ |
|:---:|:---:|:---:|
| Overlap Ratio | 32.62 | 21.45 |
| CD Dis | 32.67 | 21.98 |
| Center Dis | 32.84 | 22.39 |

Table 7: Ablation for different principles to find the closest instance for class label assignment.

The above mapping operation is also used for the coloring scheme in Fig. 5 and Fig. 9. In these figures, each color represents a unique geometric-informative prototype. Here we will give a brief introduction since readers may have questions about how to paint points with prototype-specific colors. Please refer to Fig. 2. In GP-VAE, the $N_z$ sampled features reconstruct a point cloud scene, meanwhile are used to cluster object-level features to reconstruct multiple objects. As for object coloring (Fig. 9 and Fig. 5 top), we can reversely map each reconstructed object to features $\{z_i\}$, then map each $z_i$ to points in $x$. We use the Euclidean distance to obtain the nearest prototype $geo_i$ for each $z_i$, then paint its related points with different colors. Scene coloring (Fig. 5 bottom) follows the same procedures.

## B.3 Dataset Details

Please the detailed statistics in Tab. 8 and Tab. 9, where we count per-category instance numbers in both training and testing sets of the two benchmarks. We use 'K' to denote the novel-class samples.

| Class | Split1 Train | Split2 Train | Test |
|---|---|---|---|
| Cabinet | 1427 | 1427 | 372 |
| Bed | 307 | K | 81 |
| Chair | 4357 | 4357 | 1368 |
| Sofa | K | 406 | 97 |
| Table | 1271 | K | 350 |
| Door | 2026 | K | 467 |
| Window | K | 928 | 282 |
| Bookshelf | K | 300 | 77 |
| Picture | 661 | 661 | 222 |
| Counter | 216 | K | 52 |
| Desk | 551 | K | 127 |
| Curtain | 292 | 292 | 67 |
| Refrigerator | 186 | 186 | 57 |
| Showercurtain | 116 | K | 28 |
| Toilet | K | 201 | 58 |
| Sink | 390 | 390 | 98 |
| Bathtub | K | 113 | 31 |
| Garbagebin | K | 1985 | 530 |

Table 8: Data statistics of **FS-ScanNet**.

| Class | Train | Test |
|---|---|---|
| Bed | K | 515 |
| Table | K | 2348 |
| Sofa | 706 | 627 |
| Chair | 9278 | 10016 |
| Toilet | K | 145 |
| Desk | 933 | 1882 |
| Dresser | 182 | 218 |
| Night Stand | K | 255 |
| Bookshelf | 204 | 282 |
| Bathtub | 67 | 49 |

Table 9: Data statistics of **FS-ScanNet**.

## C    Supplementary Results

### C.1    Visual Comparisons on FS-SUNRGBD Benchmark

Illustrated in Fig. 8, comparing with SOTA [1], our results are more accurate on both box center locations and box dimensions. Our predictions gain better overlap w.r.t. ground truths, and meanwhile, we achieve higher recall value with little false positive.

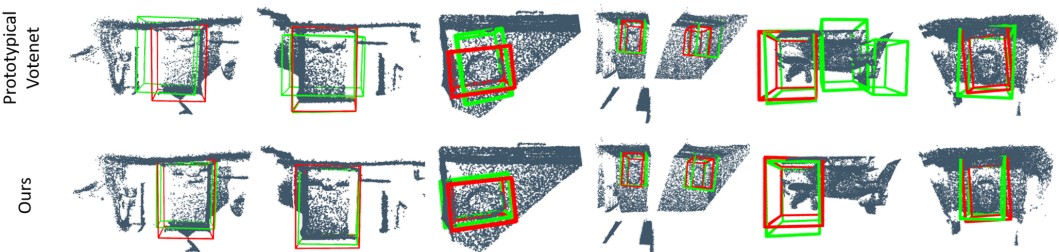

Figure 8: Exampled detection results by our method and Prototypical VoteNet (SOTA) on **FS-SUNRGBD** with $K = 5$. The red bounding boxes are the ground truths, while the green bounding boxes are the predictions.

## C.2 Comparison Results on Imbalanced Dataset Splits

To further challenge the FS3D task, an intuitive point is the imbalanced problem, i.e., imitating the long-tail settings to see how existing methods perform on higher levels of training data imbalance. Thanks to [6, 1], we can evaluate the imbalance factor of a dataset by: (i) Sort the categories by the sample counts in descending order; (ii) Reduce the training sample of each class to $N_i \times \mu^i$, where $i$ is the descending index and $N_i$ is the original data number of category $i$; and (iii) Divide the number of training samples in the smallest class into the largest. For **ScanNet v2** Benchmark [7], we first calculate the base imbalance factor **P**, then follow the above procedures to produce challenging long-tail **ScanNet v2** datasets with higher imbalance rates of **10P**, **20P** and **50P**.

Tab. 10 summarizes the quantitative comparison results. From **P** to **50P**, our results show consistent precision improvements, and for the most challenging **50P** we achieve an $+3\%$ outperformance over SOTA [1] on the $AP25$ metric. These preliminary results demonstrate that although our method suffers from unsurprising performance drops when the data imbalance factor increases, the overall performance is tolerable and in accord with our expectation for an FS3D approach.

| Method | P | | 10P | | 25P | | 50P | |
|---|---|---|---|---|---|---|---|---|
| | $AP_{25}$ | $AP_{50}$ | $AP_{25}$ | $AP_{50}$ | $AP_{25}$ | $AP_{50}$ | $AP_{25}$ | $AP_{50}$ |
| VoteNet | 62.34 | 40.82 | 52.06 | 35.64 | 43.12 | 27.13 | 40.01 | 26.77 |
| Prototypical VoteNet | 62.59 | 41.25 | 52.60 | 36.87 | 44.53 | 29.17 | 41.99 | 29.01 |
| Ours | 62.75 | 42.54 | 53.32 | 37.46 | 44.75 | 29.95 | 45.14 | 30.65 |

Table 10: Comparison results under different imbalanced splits of **ScanNet v2**.

## C.3 Comparison Results with 2DFSL-enhanced Detection Networks

2DFSL has been widely studied on various image vision tasks, in this case, incorporating the related techniques for 3D point cloud detection is worth trying. We implement three most recent 2DFSL methods (DeFRCN [8] and FSCE [2]) on two SOTA fully-supervised point cloud detection methods (CAGroup3D [9] and FCAF3D [10]), such that the original detection networks can adapt the FS3D task. Tab. 11 shows the results for **FS-ScaneNet Split-1**.

| Method | 3-shot | | 5-shot | |
|---|---|---|---|---|
| | $AP_{25}$ | $AP_{50}$ | $AP_{25}$ | $AP_{50}$ |
| CAGroup3D + DeFRCN | 3.25 | 1.08 | 4.76 | 2.31 |
| CAGroup3D + FSCE | 4.01 | 1.57 | 4.95 | 3.32 |
| FCAF3D + DeFRCN | 5.51 | 3.12 | 8.24 | 3.93 |
| FCAF3D + FSCE | 6.13 | 4.31 | 8.55 | 4.89 |
| Ours | 31.60 | 19.37 | 32.84 | 22.39 |

Table 11: Comparison results with 2DFSL-enhanced 3D point cloud detection networks.

## C.4 Ablation Studies on Different Numbers of Geometric-informative Prototypes

We ablate different numbers of geometric-informative prototypes, which is $N_{geo}$ in the main paper. Please refer to Tab. 12 and Fig. 9, the experiments are conducted on **FS-ScaneNet Split-1** 5-shot. When the number is very small, e.g., 16, the quantitative performance suffers drastic drops. This can be visually proved in the figure (the first column), we fail to construct an object by prototypes combination. It is infeasible to use only 16 prototypes to conclude all the 3D physics structures in the real world, insufficient prototype numbers can lead to severe geometric information loss of the learned prototypes. In Fig. 9 from left to right, with increasing prototype numbers, an object can be more detailed described with diverse prototypes that each specifies a unique geometric component. For indoor scenes, 128 is an appropriate number for geometric-information prototypes, this is also in accord with the discovery by Zhao et. al [1]. Simply adding more prototypes will not further improve the network performance, but rather brings redundant memory and calculation costs.

## C.5 Results on Detection Networks with Plug-in P-VAE

A good technique for FS3D should easily adapt to other point cloud detection networks for broader usages. Our P-VAE can be a simple plug-in module to construct a probabilistic latent space, it can be

| Num. of Geo-Proto | $AP_{25}$ | $AP_{50}$ |
|---|---|---|
| $N_{geo} = 16$ | 22.19 | 6.48 |
| $N_{geo} = 32$ | 28.59 | 14.52 |
| $N_{geo} = 64$ | 31.78 | 18.51 |
| $N_{geo} = 128$ | 32.84 | 22.39 |
| $N_{geo} = 256$ | 32.06 | 20.30 |

Table 12: Ablation for $N$

16 protos    32 protos    64 protos    128 protos    256 protos

Figure 9: Visualize how geometric-informative prototypes can construct objects with different prototype numbers.

implemented on various networks with different architectures. In the main paper, our P-VAE is built upon VoteNet []. Here, we utilize P-VAE as a prototype-learning feature regulator on three recent networks designed for indoor scenes: (i) transformer-based 3DETR [11]; (ii) two-stage coarse-to-fine CAGroup3D [9]; and (iii) anchor-free FCAF3D [10].

Compared with CAGroup3D which has explicit voting operations such that we can similarly adopt GP- and CP-VAE, 3DETR, and FCAF3D have no clustering module, but they still have the basic encoder-decoder architectures to apply P-VAE. In Tab. 13, equipped with P-VAE, all the methods gain consistent precision improvements on all the few-shot settings of **FS-ScanNet Split-1**, demonstrating that P-VAE is a general solution for FS3D and has wide prospects for further usages.

| Method | 1-shot | | 3-shot | | 5-shot | |
|---|---|---|---|---|---|---|
| | $AP_{25}$ | $AP_{50}$ | $AP_{25}$ | $AP_{50}$ | $AP_{25}$ | $AP_{50}$ |
| 3DETR | 7.56 | 3.16 | 10.99 | 5.29 | 18.06 | 8.98 |
| 3DETR + P-VAE | 7.91 | 3.22 | 18.47 | 9.64 | 24.16 | 10.98 |
| CAGroup3D | 1.19 | 0.16 | 3.22 | 0.96 | 4.33 | 2.06 |
| CAGroup3D + P-VAE | 3.12 | 1.45 | 5.97 | 2.33 | 7.08 | 3.49 |
| FCAF3D | 4.12 | 2.09 | 6.05 | 3.30 | 12.78 | 5.85 |
| FCAF3D + P-VAE | 4.68 | 2.15 | 6.96 | 3.88 | 13.72 | 5.93 |

Table 13: Enhance recent point cloud detection networks with our pluggable P-VAE.

## C.6  Class-specific Results

Tab. 14c summarizes the precision values of each class on three benchmarks. As we discussed before, the results have a large variance between different classes. We can observe that objects of irregular shapes and less geometric distinctiveness can have severe performance drops, e.g., windows and garbage bins. For these challenging classes, their original 3D inputs essentially lack geometric characteristics. In this case, dedicated design on point cloud networks can still result in unsatisfactory results. A feasible solution is to incorporate corresponding 2D images, for example, to adopt a teacher-student knowledge distillation framework.

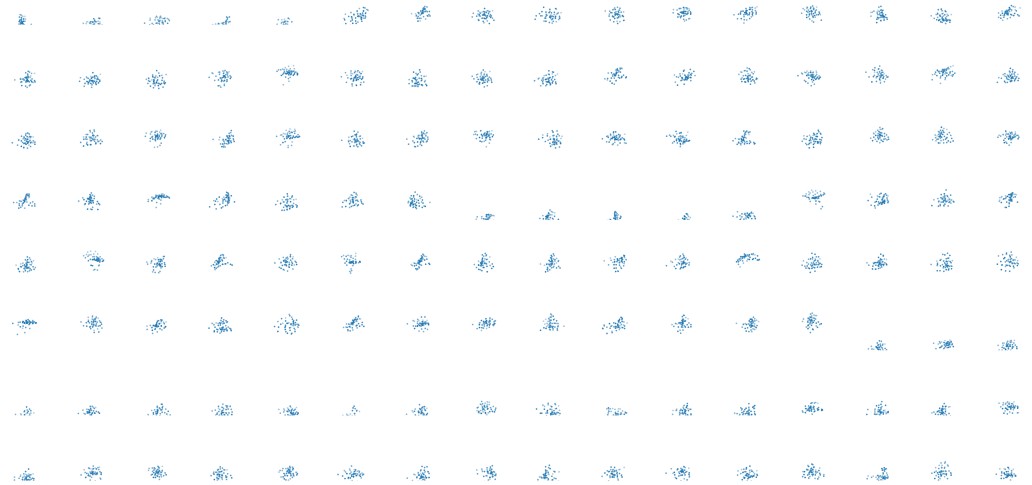

Figure 10: Complete set of 128 geometric-informative prototypes.

| Category | $AP_{50}$ | $AP_{25}$ |
|---|---|---|
| Bathtub | 50.92 | 66.36 |
| Toilet | 44.01 | 55.70 |
| Bookshelf | 1.14 | 2.48 |
| Sofa | 37.53 | 59.57 |
| Window | 0.56 | 1.94 |
| Garbagebin | 1.66 | 3.64 |

(a) Statistics of per-category results on **FS-ScanNet Split-1** 5-shot.

| Category | $AP_{50}$ | $AP_{25}$ |
|---|---|---|
| Showercurtain | 2.31 | 30.37 |
| Counter | 0.22 | 8.81 |
| Bed | 64.60 | 73.68 |
| Desk | 12.89 | 27.54 |
| Table | 8.91 | 25.85 |
| Door | 2.11 | 8.37 |

(b) Statistics of per-category results on **FS-ScanNet Split-2** 5-shot.

| Category | $AP_{50}$ | $AP_{25}$ |
|---|---|---|
| Table | 2.43 | 14.85 |
| Bed | 16.09 | 54.57 |
| Night Stand | 0.10 | 1.57 |
| Toilet | 20.00 | 56.91 |

(c) Statistics of per-category results on **FS-SUNRGBD** 5-shot.

Table 14: Statistics of per-category results on three benchmarks.

## C.7 Complete Geometric-Prototypes Visuaization

Illustrated in Fig. 10, we show all the 128 geometric-informative prototypes screen-shoot of the same camera pose.