# OpenReview forum: "Prototypical Variational Autoencoder for 3D Few-shot Object Detection"
_NeurIPS.cc/2023/Conference — NeurIPS 2023 poster_

### Official Review · Reviewer_B5ky · 2023-06-24

**Soundness:** 2 fair
**Presentation:** 2 fair
**Contribution:** 3 good
**Rating:** 6
**Confidence:** 4

**Summary:**

This paper proposes a novel approach for few-shot 3D object detection by combining prototype learning and variational autoencoders. To address the weak geometry regularization and data imbalance issues of the existing methods, it proposes a novel VAE specifically designed for prototype learning named Prototypical VAE (P-VAE). Moreover, it extends P-VAE with geometric-informative prototypes and class-specific prototypes to enhance the object detection performance in the few-shot regime. Experimental results on several benchmarks validate the effectiveness of the proposed method.

**Strengths:**

1. This paper proposes two prototypical VAE based on geometric-informative and class-specific prototypes, which helps solve the weak geometry regularization and data imbalance issues encountered by the existing few-show 3D object detection methods based on prototype learning. This combination of prototype learning and VAE is novel and effective.

2. Experimental results showed strong support for the effectiveness of the proposed method in the few-shot regime. In particular, it shows consistent improvement over baseline methods on various settings of different benchmarks.

3. The paper is well written, with easy-to-follow equations and proper notations.

**Weaknesses:**

1. In order to detect objects from the scene point clouds, a number of instances should be predicted. However, it is not clear how to obtain the number of the predicted instances $N_{ins}$ from $N_z$ features $z'_i$.

2. There is no ablation study of the two feature calibration described in Eq-6 and Eq-9, how much do they contribute to the final performance improvement?

3. What are the object detection heads in Figure-2? How are they trained exactly?

**Questions:**

Please refer to the weaknesses.

**Limitations:**

The limitations of the proposed method and potential solutions are discussed in the final section of the paper.

---

> ### Author Rebuttal · Authors · 2023-08-09
>
> We thank the reviewer for reading our submission carefully and for the insightful suggestions. We address the reviewer’s comments below. Also, we will release code upon the publication of this work.
>
> **Q1: How to obtain the number of instances $N_{ins}$?**
>
> A1: Given the per-point features $\\{z'_i\\} _{i=1}^{N_z}$, we use MLP to predict both feature offsets ($D$-dimension) and point offsets (3-dimension). We add the point offsets to the original point to get the shifted points, then adopt Furthest Point Sampling (FPS) to obtain $N_{ins}$ votes. $N_{ins}$ is produced by the network on-the-fly, depending on the input scenario. The above procedure is a clustering module following VoteNet[1], here we link pseudocode    https://drive.google.com/file/d/1f_nBjNV_eWK90lP1Xya6ByVUszC97yNw/view?usp=sharing for your reference.
>
> **Q2: Ablation studies for feature calibration.**
>
> A2: Thank you for pointing this out. In GP- and CP-VAE, we use a cross-attention module (Eq. 6 and Eq. 9) to enrich the features with the prototype information. Our studies for calibration are inadequate in the submitted version. Here we add comparisons with three calibration approaches: concatenation[2], perturbed attention[3] and adaptive attention[4]. See Table A, we find that adaptive attention can achieve better performance than base attention. The reason may be, the soft attention with meta-reweighting strategy can localize and highlight the region of interest in query samples. We will modify our method carefully and thanks again for your suggestion.
> |                          |           |           |           |           |
> |:------------------------:|:---------:|:---------:|:---------:|:---------:|
> |                          | 3-shot    | 3-shot    | 5-shot    | 5-shot    |
> |                          | AP25      | AP50      | AP25      | AP50      |
> | P-VAE                    | 31.60     | 19.37     | 32.84     | 22.39     |
> | Concatenation\[2\]       | 23.44     | 11.07     | 29.18     | 17.92     |
> | Perturbed Attention\[3\] | 29.31     | 10.09     | 31.86     | 19.23     |
> | Adaptive Attention\[4\]  | **31.89** | **20.04** | **33.01** | **22.78** |
>
> Table A: Ablations studies for feature calibration strategies on FS-ScanNet Split-1.
>
> **Q3: Detail the detection heads. Besides, how to train the heads?**
>
> A3: Thank you for pointing this out. Referring to the 'Object Detection Heads' in Figure 2 of the paper, the detection includes three steps:
>
> (i) Normalization. Given $N_{ins}$ clusters, each cluster is denoted as $C = \\{c_i\\}_{i=1}^n$, where $n$ is the number of features assigned to $C$ through FPS (discussed in A1). $c_i$ is composed of two attributes: the $D$-dimension features $w'_i$  (see Figure 2 in the paper) and the 3-dimension shifted point coordination $p_i$. We calculate the clustering center $\\bar p=\\frac{1}{n} \\sum_{p=1}^n {p_i}$ then locally normalize $p_i$ to $p'_i=(p_i - \\bar p)/r$, where $r$ is the maximum distance to the center.
>
> (ii) We concatenate $w'_i$ with $p'_i$ for the ($D$+3)-dimension $c'_i$, then use a 1-layer MLP on each $c'_i$, resulting in $n$ features. We collect them and use MaxPooling to get a single ($D$+3)-dimension feature $c$.
>
> (iii) We use a 1-layer MLP on $c$ to obtain the final vector, which contains an objectness score, bounding box information (i.e., xyzlwh) and classification scores.
>
> Similar to VoteNet, we use the standard 3D detection loss to supervise the head. We assign predictions within 0.3m to a GT as 'positive', and assign predictions far from any GT more than 0.6m as 'negative'. We use Cross-Entropy loss to supervise the pos/neg objectness. For each positive candidate, we take the nearest GT as its paired GT, which is used to supervise the box location and dimension with Smooth-L1 loss. We also use Cross-Entropy loss to supervise the classification score of each positive candidate.
>
> We will add this part in the Implementation Details section of the revised version for clarification.
>
> [1] Qi, Charles R., et al. "Deep Hough Voting for 3D Object Detection in Point Clouds." ICCV 2019.
>
> [2] He, Shuting, et al. "Prototype Adaption and Projection for Few-and Zero-Shot 3D Point Cloud Semantic Segmentation." TIP 2023.
>
> [3] Lu, Yu, et al. "Attention Calibration for Transformer in Neural Machine Translation." ACL-IJCNLP 2021.
>
> [4] Jiang, Zihang, et al. "Few-Shot Classification via Adaptive Attention." arXiv preprint arXiv:2008.02465 (2020).

---

> > ### Comment · Reviewer_B5ky · 2023-08-17
> > **Post-rebuttal comment**
> >
> > I appreciate the author's feedback, which addressed my issues. Considering the general positive comments of other reviewers, I thus keep my original rating.

---

### Official Review · Reviewer_cCbd · 2023-07-05

**Soundness:** 4 excellent
**Presentation:** 3 good
**Contribution:** 3 good
**Rating:** 4
**Confidence:** 4

**Summary:**

This paper studies a challenge task called FS3D. They first presents that the previous work on FS3D lacks fine-level supervision, as the intermediate features are simply averaged to update the prototypes, which are then used to augment features for sequential detection. In order to solve this problem, they leverage Prototypical Variational Autoencoder to conduct regularization on both local and global levels.

**Strengths:**

1. The paper is well written and easy to follow since the motivation is clear.
2. The experimental improvement is significant.


**Weaknesses:**

1. The related works on FSL for 3D Point Cloud is insufficient.
2. why does Prototypical Variational Autoencoder work in this task ? Have you tried other models for reconstruction ?





**Questions:**

1. Please see the weakness
2. Can the proposed method be used to other few-shot learning tasks in point cloud, such as segmentation ?
My main concern is that why does author choose Prototypical Variational Autoencoder as the reconstruction model. Does it have outstanding property for FS3D ?  It is the key for the acceptance of this paper.
if authors could address my concerns, I am willing to increase the score.

**Limitations:**

yes

---

> ### Author Rebuttal · Authors · 2023-08-09
>
> We thank the reviewer for reading our submission and for the insightful comments. We address the reviewer’s comments below.
>
> **Q1: Survey on 3D FSL.**
>
> A1: Thank you for pointing this out. Our work focuses on FS3D, we believe a more careful review on FSL for point cloud can lead to high-level insights. We categorize recent works into four families according to the FSL strategies. [10,14,16] incorporate self-supervision or semi-supervision. [6,7,8,9,13] regularize and align the latent space with the concept ‘prototype’. High-dimensional prototypes are used to cluster the embeddings while low-level prototypes represent basic geometric structures. [1,4,7,8,9,12,17] aim to evaluate the difference/similarity between base and novel pairs, such that the features can be separated more discriminatively. [2,3,5,11,15] enrich the limited 3D samples with data augmentation and multi-modality fusion. We will give a more detailed and comprehensive overview in the revised version.
> Here we link the reference [1-17] https://drive.google.com/file/d/1WyLnRVP53627-qZ861wJpZgq241Slefi/view?usp=sharing.
>
> **Q2: Why does P-VAE work? Why it is a better reconstruction model in comparison with other reconstruction approches?**
>
> A2: Thank you for your careful revision. Parallel to the detection task, we add a reconstruction task for preserving 3D information in the latent space. This is a widely-used self-supervised learning scheme for 3D FSL. Compared with the original reconstruction model AE, P-VAE learns a multi-center distribution that each centers at a prototype, enabling us to sample from the probabilistic space. This design imitates the real 3D environments (a prototype can have lots of variants), and also tackles the data imbalance problem (although the novel samples are limited, we can augment features by sampling). See Table A for the quantitative results of P-VAE superior to AE. In fact, the key is to obtain a parameterized latent space in the reconstruction model. Inspired by your suggestion, we study reconstruction models[18,19]. In Table B, ‘w/’ and ‘w/o’ mean whether we use variational probabilistic learning or not. We modify a CP-VoteNet as our baseline. We can find that the extra reconstruction task gains only marginal improvement (regardless of the approach), whereas the incorporation of a parameterized feature space does matter.
> For more discussion on how P-VAE helps FS3D, due to space limit please refer to this link https://drive.google.com/file/d/1EBdjNVKS194HKfJtCOCje4UPQ-fKj1ry/view?usp=sharing. We appreciate your patient reading.
> |                                 |           |           |
> |:-------------------------------:|:---------:|:---------:|
> |                                 | AP25      | AP50      |
> | Prototypical VoteNet + AE | 14.97     | 9.32      |
> | P-VAE                           | **16.00** | **10.22** |
>
> Table A: Performance of using P-VAE over [13] + AE on FS-ScanNet Split-1 (1-shot).
>
> |                        |           |           |
> |:----------------------:|:---------:|:---------:|
> |                        | AP25      | AP50      |
> | Baseline               | 13.47     | 8.33      |
> | \+ GAN\[18\] w/o       | 14.13     | 8.92      |
> | \+ GAN\[18\] w/        | 14.96     | 9.51      |
> | \+ Diffusion\[19\] w/o | 14.20     | 9.05      |
> | \+ Diffusion\[19\] w/   | 15.43     | 9.68      |
> | P-VAE                  | **16.00** | **10.22** |
>
> Table B: Reconstruction models on FS-ScanNet Split-1 (1-shot).
>
> For GAN, we pass the voted instance-level features through an MLP to predict distribution parameters, then feed the sampled features into a generator[20] to reconstruct the object, while the discriminator produces per-point scores to distinguish our prediction and GT. For diffusion model, we follow the similar scheme to incorporate probabilistic modeling, and the shape latent is replaced by each voted feature.
>
> [18] Goodfellow, Ian, et al. "Generative Adversarial Nets." NIPS 2014.
>
> [19] Luo, Shitong, and Wei Hu. "Diffusion Probabilistic Models for 3D Point Cloud Generation." CVPR 2021.
>
> [20] Li, Chun-Liang, et al. "Point Cloud GAN." arXiv preprint arXiv:1810.05795 (2018).
>
> **Q3: Does P-VAE work for other FSL 3D tasks? e.g., Segmentation.**
>
> A3: Yes. P-VAE is a general prototype learning scheme that can be plugged into many point cloud networks. Please refer to Supplementary C.5, where we can easily adapt P-VAE for various architectures.
> [21] and [22] are grouping-based instance segmentation methods. We can deploy GP-VAE at the pre-grouping stage and CP-VAE at the post-grouping stage. [23] is a transformer-based network, we then deploy GP-VAE prior to the transformer decoder, and CP-VAE between the decoder and the mask module. Considering the computational cost, P-VAE is not preferred in the shallow layers.
> We provide some preliminary results of [21] + plug-in P-VAE, comparing with a recent segmenter[24] on ScanNetv2. Please see Table C and Figure A https://drive.google.com/file/d/1fKREOe6VjEZmwRqWmLVxxRzFLpU8Rugs/view?usp=sharing. We also observe that our recall, especially for objects with larger sizes, is lower than [24]. The coarse implementation has not been thoroughly checked so we will finalize the results in the revised version.
> |                       |          |          |
> |:---------------------:|:--------:|:--------:|
> |                       | mAP      | AP50     |
> | Geodesic-Former\[24\] | **10.6** | **19.8** |
> | \[21\] + P-VAE                 | 8.04     | 15.35    |
>
> Table C: Both methods are trained on Fold0 and tested on Fold1(1-shot).
>
> [21] Jiang, Li, et al. "Pointgroup: Dual-set point grouping for 3d instance segmentation." CVPR 2020.
>
> [22] Vu, Thang, et al. "Softgroup for 3d instance segmentation on point clouds." CVPR 2022.
>
> [23] Schult, Jonas, et al. "Mask3D: Mask Transformer for 3D Semantic Instance Segmentation." ICRA 2023.
>
> [24] Ngo, Tuan, and Khoi Nguyen. "Geodesic-Former: A Geodesic-Guided Few-Shot 3D Point Cloud Instance Segmenter." ECCV 2022.

---

### Official Review · Reviewer_WEFx · 2023-07-05

**Soundness:** 2 fair
**Presentation:** 3 good
**Contribution:** 2 fair
**Rating:** 5
**Confidence:** 3

**Summary:**

 The paper proposes an approach to enhance Few-Shot 3D Point Cloud Detection (FS3D) through prototype learning with VAEs. The authors leverage VAEs to learn prototypes represented by GMM-like distributions. Two VAEs are specifically designed to preserve geometric information and refine instance features. The effectiveness of the approach is validated through various experiments.


**Strengths:**

 1.The paper introduces an approach utilizing VAEs to learn geometric and class-specific prototypes with improved performance on FS3D tasks.

 2.The effectiveness of the modules and prototypes is demonstrated through various experiments.

 3.The writing is easy to follow.

**Weaknesses:**

 1.The authors do not provide a comprehensive and detailed discussion of their motivations for the proposed approach. This should include a clear identification of the core issues, a thorough evaluation for the limitations of previous works in addressing this problem, and a convincing explanation of how incorporating VAEs can effectively tackle these limitations. It would greatly enhance the paper's credibility if qualitative and quantitative results were presented to support their motivations and demonstrate the superiority of the proposed method.

 2.The sensitivity of the number of geometric-informative prototypes as observed in Table 6 raises concerns about the practical applicability of the proposed approach. It would be beneficial if the authors discussed the implications of this sensitivity and provided insights on how to determine an optimal or adaptive number of Geo-proto for real-world applications.


**Questions:**

 1.The paper does not explain the reason why prototypes learned by previous methods lose substantial 3D information and become less geometric-informative.

 2.The motivation for using VAEs is not fully stated.

 3.Are there any empirical results to demonstrate the underrepresentation / overfitting problem of prototypes?

 4.Why can GMM-like distribution handle the overfitting problem of the latent space?

**Limitations:**

 1.The authors have addressed the limitations that data-rooted problem will lead the representative prototypes to distort and perturb the latent space negatively.

 2.Additionally, the proposed approach depends on the number of prototypes. The sensitivity raises concerns about the practical applicability of the proposed approach.

---

> ### Author Rebuttal · Authors · 2023-08-09
>
> We thank the reviewer for taking the time to read our submission and for the valuable comments. We address the reviewer’s comments below.
>
> **Q1: Why are the prototypes learnt by previous methods less geometric-informative? More detailed discussions about the limitations of previous works.**
>
> A1: Thanks for your in-depth comments. The typical FS3D methods related to geometric prototypes are [1][2]. Since [2] do not explicitly learn the prototypes, we will mainly discuss [1] and briefly mention [2].
>
> [1] learns geometric prototypes in the encoder embedding space, this high-dimensional disordered space (where they group features and use average each group as one prototype) is directly connected to the down-stream task. The framework is supervised by only a task loss, which is specified to fit the target GT (i.e., 3D boxes). Without extra regularization, the deep network loses control of the latent space. The linked Figure A {https://drive.google.com/file/d/1SVDLQj6F0wXsO-kO0-HuYg-0ERVaCkNW/view?usp=sharing} illustrates how geometric prototypes can construct a whole scenario (please see Supplementary B.2 for implementation details). Each prototype is painted with a unique color. Figure A demonstrates that the latent space of [1] fails to preserve meaningful 3D information. The geometric prototypes are not well-organized and show high randomness. In contrast, our prototypes retain reasonable physics structures and semantic consistency.
>
> The prototypes in [2] are not learnable features but 3D point clouds. These low-level prototypes are generated from random points through affine transformations, then clustered into categories by paired similarity. Please see Figure B https://drive.google.com/file/d/1xTdfyuL8mtXrlGED2-lsaQWBUqFDFB-x/view?usp=sharing , such prototypes can be very unrealistic, based on our common sense for real-world objects.
>
> **Q2: The motivation of using VAE.**
>
> A2: The core of FS3D is understanding 3D scenes with the least data requirement. We detail the motivation from four aspects.
>
> (i) For Geometric-informative Prototypes (GP), we use the set of a fixed number of GP to describe the entire 3D world, thus we expect each GP can have different shape variants given different contexts (e.g., the 'stick' GP can be short in a chair but long in a desk). VAE allows us to sample diverse features based on the prototypes (Gaussian centers), thus can provide a wide range of variants.
>
> (ii) For Class-specific Prototypes (CP), the base-class prototypes can be superior to the novel-class ones because of data imbalance, shown in Figure C of A3. We therefore use VAE for alignment. The parameterized latent space can help augment data through covering various features around a prototype center.
>
> (iii) Particularly for the novel classes CP, since we access very limited samples, we cannot fully understand all the shape variants of a category, as shown in Figure D https://drive.google.com/file/d/1k0bP29YyCBvvErqhcCCSZQ5S3q-bz-NB/view?usp=sharing. The distribution space of VAE retains higher feature transferability, such that we can learn class-specific prototypes instead of instance-specific ones.
>
> (iv) VAE is a generative model, its reconstruction task can be a regularization scheme to preserve 3D information of the latent space, as we discussed in A1.
>
> **Q3: Results that can explain the underrepresentation/overfitting of prototypes.**
>
> A3: Please see Figure C https://drive.google.com/file/d/13xzTPOpgCTA_FqKbGgA6C0AWD6YuhpEJ/view?usp=sharing for comparison results. Due to the data imbalance problem, for the baseline[1], we can observe that the quality of novel-class prototypes is significantly worse than that of base classes, whereas P-VAE can generate equally convincing prototypes without much bias.
> As an extension for A2 (iii), we conduct the following experiments. Given the novel class ‘Table’, we split the training samples into two sets w.r.t. their shapes: Standard Tables (i.e., the four-leg tables with a rectangular tabletop, as we can commonly find in the real world), and; Strange Tables (i.e., ones with a round or irregular-shaped top, supported by a steady base instead of four sticks). The networks are trained on one type and tested on another. The quantitative results are in Table A. [1] fits the given instance and cannot generalize for all tables, whereas P-VAE learns the generalized category information thus can better transfer to other table-like shapes.
> |                           |                                 |                                 |                                 |                                 |
> |:-------------------------:|:-------------------------------:|:-------------------------------:|:-------------------------------:|:-------------------------------:|
> |                           | Train on Table1, Test on Table2 | Train on Table1, Test on Table2 | Train on Table2, Test on Table1 | Train on Table2, Test on Table1 |
> |                           | AP25                            | AP50                            | AP25                            | AP50                            |
> | Prototypical VoteNet\[1\] | 18.68                           | 7.01                            | 16.92                           | 6.72                            |
> | P-VAE                     | **23.05**                       | **10.44**                       | **21.97**                       | **9.41**                        |
>
> Table A: Using Standard-shaped table (Table1) and Strange-shaped table (Table2) for training/testing on FS-ScanNet Split-2 (5-shot).
>
> **Q4: Why can the GMM-like distribution handle the overfitting problem?** & **Q5: More discussion on the number of prototypes (related to Table 6 in the paper). How sensitive is it? How to determine an optimal number?**
>
> A4 & A5 & Reference: Due to space limitation, we link the response here https://drive.google.com/file/d/1FVx8bJ6zxUTpsBfvExtRuwnaF4MvbOhH/view?usp=sharing. We appreciate your patient reading.

---

> > ### Comment · Reviewer_WEFx · 2023-08-17
> >
> > Thanks for the authors' efforts and responses. However, I remain unconvinced by the explanations provided for Q2 and Q4, as they do not offer additional insights. Therefore, I will maintain my rating as borderline accept.

---

### Official Review · Reviewer_dL8b · 2023-07-06

**Soundness:** 2 fair
**Presentation:** 2 fair
**Contribution:** 3 good
**Rating:** 6
**Confidence:** 4

**Summary:**

The paper introduces Prototypical Variational Autoencoder (P-VAE) for Few-Shot 3D Point Cloud Object Detection. It tackles the preservation of geometric information and data imbalance through learning distribution parameters. The authors propose two extensions, GP-VAE and CP-VAE, focusing on geometry and class specificity. Experiments demonstrate improved performance over state-of-the-art methods.

**Strengths:**

1. The P-VAE is innovative, focusing on learning distribution parameters instead of features, which is useful for few-shot learning.

2. The encoder-decoder architecture effectively preserves geometric information, critical for 3D detection.

3. The paper includes a robust experimental evaluation, showing performance gains and insights into prototype contributions.

**Weaknesses:**

1. The paper does not include experiments to assess the effectiveness and generalization of P-VAE across different domains on few shot learning, despite P-VAE being one of the main contributions.

2. The observed improvements in the main experiments are relatively marginal, particularly in regard to the evaluation metric AP25.

**Questions:**

None

---

> ### Author Rebuttal · Authors · 2023-08-09
>
> We thank the reviewer for taking the time and reviewing our paper. Overall, most of the review comments request clarifications and minor revision on the paper. We will carefully revise the paper accordingly.  Also, we will release code upon the publication of this work.
>
> **Q1: Effectiveness and generalization of P-VAE across different datasets.**
>
> A1: Thanks for your insightful suggestion. Cross-dataset validation is indeed critical to address the distribution gap between point cloud data, which can be commonly observed in the real world. Inspired by your suggestion, we conduct a series of experiments across the FS-ScanNet and FS-SUNRGBD datasets. We compare the AP results on not only VoteNet [1] (the vanilla detector) and Prototypical VoteNet [2] (the few-shot learning detector) but also an implemented transferable P-VAE+. Since Cross-domain Few-shot learning (CDFS) for point cloud object detection is under-explored, we adapt a 2D CDFS method [3], which utilizes high-order self-supervision to regularize the latent space, the training scheme follows [4]. Please see Table A-C below for the details.
>
> - Part 1 Train on FS-ScanNet Split-1 then test on FS-SUNRGBD (5-shot).
> | |AP25|AP50|
> |:---:|:---:|:---:|
> |VoteNet[1] + FineTune|20.66|1.64|
> |Prototypical VoteNet[2]|21.95|1.77|
> |P-VAE|23.49|**1.91**|
> |P-VAE+|**23.52**|1.87|
>
> Table A: Train on base classes in FS-ScanNet split-1 (Cabinet, Bed, Chair, Sofa, Table, Door, Picture, Counter, Desk, Curtain, Refrigerator, ShowerCurtain, and Sink), then test on novel classes in FS-SUNRGBD (Toilet and Night Stand).
>
> - Part 2 Train on FS-ScanNet Split-2 then test on FS-SUNRGBD (5-shot).
> | |AP25|AP50|
> |:---:|:---:|:---:|
> |VoteNet[1] + FineTune|24.92|13.08|
> |Prototypical VoteNet[2]|26.25|18.90|
> |P-VAE|28.67|19.30|
> |P-VAE+|**29.02**|**19.76**|
>
> Table B: Train on split-2 base classes in FS-ScanNet (Cabinet, Chair, Sofa, Toilet, Door, Picture, Counter, Desk, Curtain, Refrigerator, ShowerCurtain, and Sink), then test on novel classes in FS-SUNRGBD (Bed, Table, and Night Stand).
>
> - Part 3 Train on FS-SUNRGBD then test on FS-ScanNet (5-shot).
> | |AP25|AP50|
> |:---:|:---:|:---:|
> |VoteNet[1] + FT|23.18|2.98|
> |Prototypical VoteNet[2]|25.93|3.54|
> |P-VAE|26.17|3.73|
> |P-VAE+|**26.73**|**3.76**|
>
> Table C: Train on base classes in FS-SUNRGBD (Sofa, Chair, Desk, Dresser, Bookshelf, and Bathtub), then test on novel classes in FS-ScanNet (Bed, Table, Toilet, and Garbagebin).
>
> We will survey other CDFS approaches and exploit them on point cloud object detection. In the revised version, we will provide additional cross-dataset results to support further studies on 3D CDFS.
>
> **Q2: Explanation for the marginal improvements, especially on AP25.**
>
> A2: Thank you for your revision on the experiment section. For both benchmarks, our method achieves consistently top on the quantitative results, as shown in Tables 1 and 2 in the paper.
> (1) As for those comparable ones, in fact P-VAE gains significant improvements on the common-in-real-world categories. Please see Table D. These objects frequently appear in daily life, thus their precisions are critical for practical usages. Please also see the qualitative results in the linked Figure A https://drive.google.com/file/d/1_yfH-zu4PIa-vFGEr9AZ-0FF4bcGggRO/view?usp=sharing (top: FS-ScanNet and bottom: FS-SUNRGBD), where P-VAE predicts comparable or less boxes, but with superior accuracy (similar xyz location, but better lwh dimensions).
> (2) This result of Figure A is in accord with your observation, that our improvements on AP50 are stronger than AP25. AP50 is a stricter metric. In some cases, [2] predicts a coarse box with obvious deviation, though it can be counted for AP25, it is actually a false positive for AP50.
> (3) As for your suggestion on AP25, a possible solution for producing more predicted candidates is to augment novel samples with PointMix [5] then use the post-augmentation data for self-ensembling training[6]. The implementation is still on-going so we will provide the results in the revised version.
> |               |                    |                    |                           |                           |
> |:-------------:|:------------------:|:------------------:|:-------------------------:|:-------------------------:|
> | Category      | P-VAE              | P-VAE              | Prototypical VoteNet\[2\] | Prototypical VoteNet\[2\] |
> |               | AP25               | AP50               | AP25                      | AP50                      |
> | Bathtub       | **42.99 (+9.97)**  | **35.75 (+13.71)** | 33.02                     | 22.04                     |
> | Toilet        | **45.51 (+6.61)**  | **19.31 (+3.80)**  | 38.80                     | 16.23                     |
> | ShowerCurtain | **29.14 (+13.80)** | **4.98 (+1.26)**   | 15.34                     | 3.72                      |
> | Sofa          | **14.27 (+2.31)**  | **9.81 (+3.35)**   | 11.96                     | 6.46                      |
>
> Table D: Results on common-in-real-world classes on FS-ScaNet Split-1(1-shot).
>
> [1] Qi, Charles R., et al. "Deep Hough voting for 3D Object Detection in Point Clouds." CVPR 2019.
>
> [2] Zhao, Shizhen, and Xiaojuan Qi. "Prototypical VoteNet for Few-Shot 3D Point Cloud Object Detection." NeurIPS 2022.
>
> [3] Yuan, Wang, et al. "Task-level Self-Supervision for Cross-Domain Few-Shot Learning." AAAI 2022.
>
> [4] Oh, Jaehoon, et al. "Understanding Cross-Domain Few-Shot Learning Based on Domain Similarity and Few-Shot Difficulty." NeurIPS 2022.
>
> [5] Chen, Yunlu, et al. "PointMixup: Augmentation for Point Clouds." ECCV 2020.
>
> [6] Zhao, Na, Tat-Seng Chua, and Gim Hee Lee. "SESS: Self-Ensembling Semi-Supervised 3D Object Detection." CVPR 2020.

---

> > ### Comment · Reviewer_dL8b · 2023-08-17
> >
> > I read the author's rebuttal carefully and it solves my concerns well. For question 1, although the improvement is relatively marginal, the author did conduct more experiments on different datasets to showcase the effectiveness and generalization of the proposed method. For question 2, the author clarifies that the performance improvement of AP50 is good and AP50 is a stricter metric. Therefore, I will raise my rating to weak accept.

---

### Decision · Program_Chairs · 2023-09-21

**Decision:**

Accept (poster)

**Comment:**

The paper proposes an approach to enhance Few-Shot 3D Point Cloud Detection (FS3D) through prototype learning with VAEs. The reviewers note the novelty of the approach and its good empirical performance. They raised concerns regarding the motivation of the method and its sensitivity in the number of prototypes. The rebuttal submitted by the authors addressed these concerns.
The AC read the paper and found the presentation and command of English and accuracy of expression below the bar of acceptance. Yet, the result of the paper showing that reconstruction tasks assist few-shot discriminative tasks is an interesting one, that may be useful for the community, outweighing the low presentation quality of the paper.